# Technological Interventions to Implement Prevention and Health Promotion in Cardiovascular Patients

**DOI:** 10.3390/healthcare12202055

**Published:** 2024-10-16

**Authors:** Ayisha Z. Bashir, Anji Yetman, Melissa Wehrmann

**Affiliations:** 1The Child Health Research Institute (CHRI), The University of Nebraska Medical Center (UNMC), Omaha, NE 68198, USA; ayetman@childrensnebraska.org (A.Y.); mwehrmann@childrensnebraska.org (M.W.); 2Department of Pediatrics, Division of Cardiology, The University of Nebraska Medical Center, Omaha, NE 68198, USA

**Keywords:** health promotion, cardiology, technology, cardiac rehabilitation, telemedicine and telehealth

## Abstract

**Background/Objectives**: The aim of the narrative review is to identify information on the impact of technological interventions (such as telehealth and mobile health) on the health promotion of cardiac patients from diverse populations. **Methods**: The online databases of PubMed and the Cochrane Library were searched for articles in the English language regarding technological interventions for health promotion in cardiac patients. In addition, a methodological quality control process was conducted. Exclusion was based on first reading the abstract, and then the full manuscript was scanned to confirm that the content was not related to cardiac patients and technological interventions. **Results**: In all, 11 studies were included in this review after quality control analysis. The sample size reported in these studies ranged from 12 to 1424 subjects. In eight studies mobile phones, smartphones, and apps were used as mHealth interventions with tracking and texting components; two studies used videoconferencing as a digital intervention program, while three studies focused on using physical activity trackers. **Conclusions**: This review highlights the positive aspects of patient satisfaction with the technological interventions including, but not limited to, accessibility to health care providers, sense of security, and well-being. The digital divide becomes apparent in the articles reviewed, as individuals with limited eHealth literacy and lack of technological knowledge are not motivated or able participate in these interventions. Finding methods to overcome these barriers is important and can be solved to some extent by providing the technology and technical support.

## 1. Introduction

Cardiovascular disease is responsible for nearly 30% of all deaths globally and is deemed the leading cause of death and disease worldwide [1]. Self-monitoring for signs of emerging complications and better self-care in between face-to-face health care provider office visits can contribute to improved cardiovascular outcomes in this patient population [1,2,3]. Due to the complexity of medication regimens, comorbidities, and lack of caregiver support, self-management of the patient, adequate home care, and cardiac rehabilitation (CR) are often challenging. At the same time, there is a need for health promotion and preventive measures consisting of lifestyle behavior modifications including healthy diet, increased physical activity, weight loss, and smoking cessation [1,2]. Further challenges exist due to the limited interaction of the health care providers in resource-constrained settings, such as rural areas, impoverished neighborhoods, and low-income populations [3,4,5]. The use of telecommunications technology to provide health care from a distance, also called telehealth, is very popular and is implemented in medical care for a variety of chronic diseases, including cardiovascular diseases [1,2,4]. The effective use of telehealth interventions and patient-centered mobile health technology helps extend the reach of health systems to provide continuous support to the cardiovascular patient population belonging to all age groups and demographics [4,5,6]. 

Starting in utero, cardiovascular monitoring of maternal and fetal health is essential and implemented through regular visits with the health care provider along with prenatal ultrasounds conducted at the recommended timelines [6]. The health outcomes of mothers and infants can be improved through trustworthy guidance and regular monitoring of parameters such as weight, physical activity, and nutrition. This can be achieved through mobile health (mHealth) tools such as mobile applications (apps) targeted for maternal use during pregnancy and postpartum [6,7,8]. Understanding the implementation of successful mHealth interventions in cardiovascular disease management and their use in a wide variety of populations is important. The unequal access and use of information technology based on social and geographical characteristics is generally referred to as the digital divide. This divide contributes to some of the disparities while implementing technological interventions amongst diverse communities [5,6,7]. Digital studies generally consist of adult male populations owning smartphones; however, cardiovascular disease is present in all age groups regardless of gender, race, or socioeconomic status [4,6,7,8]. Additionally, the congenital heart disease population remains underrepresented in many studies [5,6,7]. This review aims to address the limitations and explore the status of mobile health (mHealth) strategies, in cardiovascular patients in real clinical and community settings, consisting of diverse populations and demographics.

## 2. Methods

We conducted a broad search in PubMed and the Cochrane library aligned with the goals for narrative reviews. To capture the updated literature on the efficacy of these technological interventions, we limited studies to publications between 2014 and 2024. In addition, a methodological quality control process was conducted [9]. Due to heterogeneity in the nature of the studies and implementation of the technologies, we did not conduct a meta-analysis and, therefore, report results narratively.

Search strategy: The medical subject heading (MeSH) terms of “cardiovascular”, “cardiac”, “telehealth” or “telemedicine”, “health promotion”, and “technology” were used (search terms: cardiology (Cardiac* OR cardiovascular); telehealth, telemedicine; technology, and health promotion). The reference lists from the articles included were searched for additional relevant articles. 

Inclusion and exclusion criteria: The articles identified in the literature search were thoroughly reviewed in order to meet the inclusion and exclusion criteria and appropriateness for the objective of the narrative review. The articles included in this review were qualitatively synthesized to meet the requirements from the checklist [9]. The criteria for inclusion were publications (i) available in the English language; (ii) pertaining to technological interventions; and (iii) including the cardiac population (for example, telemedicine or mobile health’s role in health promotion, cardiac rehabilitation, etc.) (Table 1). This review was not limited by the study design of the articles. Exclusion was based on reading the abstract first and then going through the full manuscript to confirm that content was not related to the cardiac population and electronic technology.

Quality control: The methodological quality examination of every article was conducted using the methodological checklist “Critical review form quantitative studies” [9] (Table 2). The articles were analyzed for technological interventions in the cardiac population. The findings, limitations, and proposed solutions presented in these studies were summarized.

## 3. Results

The chosen search yielded 37 articles, which were narrowed down to 11 pieces relevant to the topic (Table 1). Articles that were unrelated to cardiology patients or other pathological conditions such as neurological or endocrine disorders, as well as those that did not include technology, were not included in this review. In general, the papers included design, sample size, and proper analysis (Table 2); therefore, we included all 11 articles in this review after a quality control analysis.

### 3.1. Descriptive Analysis

The outcomes with respect to the technological intervention type, if reported, and patient population are described in this review. Table 1 describes the content of the included studies. The studies included a variety of research methods and design strategies including, but not limited to, randomized trials and qualitative and quantitative studies. The sample size reported in these studies ranged from 12 to 1424 subjects [10,11,12,13,14,15,16,17,18,19,20]. 

### 3.2. Types of Technological Intervention and Patient Population

In eight studies mobile phones, smartphones, and apps were used as the mHealth intervention with a tracking and texting component [10,11,12,13,14,15,16,19]; two studies used videoconferencing (telehealth) as a digital intervention program [18,19]; while three studies focused on using physical activity trackers [11,17,20]. The summaries of the interventions presented in these studies and an outline of the results are described below.

Text messages and medication adherence [10]: In the study called “Text Messages to Improve Medication Adherence (TextMeds) and Secondary Prevention after Acute Coronary Syndrome (ACS) [10]”, the effects of text-message-delivered cardiac education, support, and adherence to medication schedules were studied after ACS. Providing supportive and consistent care to ACS patients after discharge remains an implementation challenge for many hospitals and health systems [10,20]. It is important to reduce the burden of readmissions and provide the post-discharge care needed through the implementation of best-practice secondary prevention strategies such as healthy living, appropriate medication, and emotional well-being [20,21]. The intervention consisted of a personalized and customized text message-based program for one year, which required minimal central staff support. Intervention participants were more motivated to eat the recommended servings of fruit and vegetables and normalize their body mass index. However, the study found that among the participants there were no significant effects on the primary outcomes of blood pressure, low-density lipoprotein cholesterol, and medication adherence. In contrast, there were high levels of acceptability of the intervention, program engagement, and usefulness in being a credible source of information and support to the patients after discharge [10].A mobile application for technology-facilitated home cardiac rehabilitation [11]: In Smart HEART (Health Education and Rehabilitation Technology), direct communication between the patient and a health coach was provided through a mobile smart phone app to encourage self-monitoring. A wrist-worn activity tracker encouraged regular exercise [11]. Remote CR enhanced with a digital health intervention (DHI) was provided to patients to check usability and to determine whether the intervention improved CR access, patient-reported outcomes, and cardiac risk factors [11]. After the in-person baseline visit, the participants received a 3-month, remote CR program that consisted of structured home exercise enhanced with the Movn smartphone app (Movn Health, Irvine, CA, USA) and a wearable fitness tracking device. The data were shared with a dedicated health coach [11,12]. The results indicated that the DHI-enhanced remote CR program was associated with enhanced CR access, improvement in the markers of cardiovascular risk, and healthy behaviors.A patient-centered digital health intervention for cardiac rehabilitation [12]: The study consisted of 258 patients enrolled in remote CR enhanced with a DHI. In order to participate in the DHI arm of the cardiac rehabilitation program at the Veteran Health Administration (VHA) or VA medical center, the participants were required to own an Android or an iOS smartphone, enabled with access to Wi-Fi or a data plan. The DHI intervention consisted of remote CR with a structured, 3-month home exercise program partnered with multi-component coaching, a commercial smartphone app, and a wearable activity tracker. Patient-reported outcomes from pre- to post-intervention were measured along with changes in 6-min walk distance, cardiovascular risk factors, and intervention completion rates. The results showed the intervention was associated with enhanced CR access, improved markers of cardiovascular risk, high completion rates, and healthy behaviors.A smartphone application after cardiac rehabilitation improves exercise capacity with long term follow-up [13]: In this study conducted in Norway, the intervention group received an app that was developed to guide and help patients change their behavior and maintain habits, after cardiac rehabilitation. A physiotherapist was the supervisor of the group for a year, and the patients could submit questions and receive feedback via email regularly as well as receiving short, tailored, motivational feedback regularly [13]. The results indicated that, compared with a control group post-CR, improvements were seen in VO2peak, exercise performance, and exercise habits, as well as self-perceived goal achievement.Mobile health and implantable cardiac devices: patients’ expectations [14]: Remote monitoring systems in patients with implantable cardioverter defibrillators (ICDs) are a common area of implementation of mHealth in clinical practice. In this study conducted in an outpatient clinic in Italy, the patients’ perspectives on and interest in receiving data from their implantable cardiac devices and clinical and health related advice via remote monitoring were studied. The ICD patients showed interest in receiving information pertaining to the technical functioning of the device, but there was a lack of interest regarding the role of these tools in self-management of the disease. The presence of caregiver support as well as higher education were associated with a greater interest in receiving information via the mobile phone [14]. These results were limited by being conducted in a single center and should be expanded to other centers to achieve an impact in the future development of novel, mHealth patient-centered devices.Participation of young African American females in an mHealth study in cardiovascular disease reduction [15]: In the study, 40 Black female participants with cardiovascular disease (CVD) risk factors, age 25 to 45 years, participated in a 4-week (two hours per week) intervention consisting of self-management educational classes and six months of wireless coaching and monitoring. The women responded to a semiqualitative online survey assessing the user-friendliness and perceived helpfulness of the intervention at follow-up.The results were favorable, with positive implications for practice. Most of the participants did not encounter barriers to participation, which suggests that mobile health interventions can be effective tools to improve health behavior patterns and provide helpful support in the prevention of cardiovascular disease. Targeting women provided indirect benefits for other family members, especially children. The women mentioned their family members were more inclined to participate in healthy habits. This study had a few limitations, including being conducted on a small sample size in urban Southern California and, therefore, cannot be generalized to other African American communities. In order to help bridge some of the disparities in access to health care in this population, a larger-scale multicenter trial would be helpful and help validate the findings of the study [15].Evaluation of the impact of the HeartHab application on motivation, physical activity, quality of life, and risk factors of patients with cardiovascular disease [16]: The aim of this Belgian study was to investigate the impact of the HeartHab application on the patients’ motivation, physical activity, exercise target fulfilment, QoL, and modifiable risk factors in those with coronary artery disease (CAD) during telerehabilitation. In total, 32 CAD patients were randomized on a 1:1 ratio to telerehabilitation or usual care. The persuasive design techniques integrated in HeartHab and the tailoring of exercise targets were effective in motivating patients to reach their telerehabilitation targets. The results demonstrated positive improvements in VO2max, glucose, HDL cholesterol, weight, and quality of life [16]. A larger sample size and longer evaluation would be beneficial and shed more light on these results.An automated mHealth intervention for physical activity promotion [17]: Smartphone users aged 18 to 69 years were enrolled in the mobile active (mActive) study at an ambulatory cardiology center in Baltimore, Maryland. In this study, smart texts through a smartphone delivered coaching three times a day aimed at individual encouragement. The participants used their own smartphones, and feedback loops were created by a fully automated, physician-written, theoretical algorithm, which used patient real-time activity data, 16 personal factors and, as a goal, 10,000 steps per day. Digital physical activity tracking was performed using a wearable, display-free, triaxial accelerometer that paired with low-energy Bluetooth and compatible smartphones. Smart texts with activity tracking led to the best physical activity outcomes, such as increased daily steps (better outcomes than tracking alone).The mHealth intervention with the smart text component and digital tracking significantly increased physical activity. Despite positive results, this was considered a pilot study. Future steps of including human coaches and increasing the sample size will be beneficial in understanding the impact of this mActive study [17].A live videoconferencing intervention in pediatric heart transplant recipients [18]: In this study, the feasibility and impact of a supervised exercise and diet intervention delivered via videoconferencing was tested at least one year after transplant in subjects recruited from one center located in the San Francisco Bay Area [18]. The lifestyle intervention in the study’s pediatric heart transplant recipients resulted in excellent adherence and improvements in cardiac, vascular, functional, and nutritional health. After transitioning to the maintenance phase, several of the health indices were sustained. The researchers aim to shift the clinical focus from “exercise restrictions to exercise prescriptions” in a vulnerable pediatric population.Physical activity trackers and pediatric patients with Marfan syndrome [19]: In this clinical intervention, 24 pediatric patients with Marfan syndrome between 8 and 19 years of age participated, and their physical activity was tracked. They were instructed to take 10,000 steps per day for 8 months. The aortic outflow and root (AoR) dimension, arterial stiffness, endothelial function, physical activity indices, inflammatory biomarkers, and coping scores were measured at baseline and at 6 months. This study demonstrated the feasibility of a physical activity intervention in pediatric patients with Marfan syndrome and the potential to decrease the AoR dilation rate [19]. The focus has been more toward exercise restrictions than toward promotion in this patient population; the researchers hope that the study results might help shift the paradigm [21]. Additional, similar studies can help provide guidelines on how supervised exercise therapy can be further explored in a multicenter study with a larger sample size.Virtual cardiac fitness training in pediatric heart transplant patients [20]: Participants between 10 and 20 years old underwent an intervention that consisted of exercise sessions twice a week for 30 min under the supervision of a trained exercise physiologist over a virtual platform for 16 weeks. The patients wore a FitBit accelerometer to monitor daily activity levels throughout the duration of the study. At the conclusion of the intervention, the participants repeated the strength and flexibility assessment, a 6MWT, and quality of life parameter measurements to compare with the baseline measurements. The results of the study showed successful implementation of the intervention with excellent adherence and improvement in physical fitness and quality of life.In general, the articles included in this review mention patient satisfaction with the technological interventions and access to health care providers [10,11,12,13,14,15,16,17,18,19,20]. These modalities provide cardiac patients with a sense of security and well-being. While these studies mention that the enrolled patients found the technological interventions easy to implement, the issue of the digital divide is voiced in the studies [10,11,12,13,14,15,16,19]. This issue is discussed in more detail in the sections below.

## 4. Discussion

The evidence based on the above literature search suggests that the implementation of technological interventions including text messages, smart phone apps, physical activity interventions, video conferencing, telehealth, and other methods mentioned in these articles is feasible and accepted by diverse cardiac patient populations [15,16,17,18,19,20]. These studies are important in laying the groundwork for improving cardiac patients’ health and providing positive reinforcements [10,11,12,13,14,15,16,17,18]. Patients with heart disease can receive proper support and education from the implementation of digital health strategies [21,22,23,24,25]. Reassuringly, cardiac patients seem to be seeking information from helpful organizations and other credible sources including health professionals such as cardiologists, primary care physicians and staff, and nursing and allied health care providers [26,27,28]. Family engagement and caregiver support are improved with these technologies, and the parents of pediatric cardiac patients find this category of methods useful and acceptable, yielding positive results [18,19,20].

The studies conducted through the Veteran Health Administration mention that larger studies consisting of a wider variety of long-duration research are needed to understand the role of their program by comparing with other centers. This step will help in creating an understanding of the impact on long-term health outcomes and hospital readmissions due to coronary diseases [12,23]. Overall, digital health interventions in the cardiac population at the VHA had positive results, despite focusing on a high-risk patient population with generally low socioeconomic status and also factoring in that many such studies are challenging to implement because of VA-specific privacy requirements.

In the study conducted by Kathuria-Prakash et al., smartphones were the main source of Internet access in the African American female population and the barriers mentioned stemmed from providing an additional phone [15]. Some participants forgot to charge the device and experienced technical difficulties while using the device in the smartphone studies [12,15,23]. These issues can be resolved by implementing minor changes to the study design, such as using a different fitness tracking modality or wearable device [15]. For example, in another study, a lithium battery was provided with the device and did not need recharging for the study duration [17]. Our results provided findings similar to other studies that show that diverse populations with heart disease benefit from mHealth and technological interventions, including those who have suboptimal access to health care resources, transportation, and often a lower health literacy level [27,28,29,30,31,32,33]. The studies mention that patients were more inclined to self-care and felt a sense of security as the health providers were continuously monitoring their situation and were aware of their health issues [20,21,34,35].

### Limitations and Future Directions

This review did not comprise the level of rigor that is implemented in systematic reviews; there are likely missed articles in the narrative review approach. However, for the articles included in this review, some of the common factors identified are that the studies were limited by access to technology, and only individuals with smartphones were enrolled in studies requiring cell phone participation by the patients [10,11,12,13,14]. Some of the studies were conducted by the VA health system, in which all the participants were veterans belonging to the integrated health care system, consisting mostly of males, which limits generalizability [11,19]. The sicker and older veterans were often not comfortable with technology and, therefore, did not enroll [11,19,23]. In the rural population, the VHA should equip the eligible veterans with devices that have Wi-Fi capabilities to increase access and participation and limit inclusion bias [36,37]. Lessons can be gleaned from the studies reviewed showcasing the novel coronavirus (COVID-19) pandemic’s impact on the field of telehealth cardiology [38,39,40,41]. While the pandemic transformed health care worldwide, studies showed advantages in those cardiac outpatient settings in which clinics adopted telehealth as the new normal [37,41,42]. COVID-19’s intersection with cardiology is a robust and wide topic that should be addressed in another paper dedicated to just this topic; hence, it is better addressed in the future.

At the same time, much can be learned from the methods implemented in pilot studies and health care systems existing outside the United States. This review includes studies conducted in Australia, Canada, Norway, Italy, and Belgium, though it may not be possible to straightforwardly replicate the interventions cited here due to differences in the health care systems and reimbursements [7,13,14,16]. Future directions should focus on embracing further studies and identifying those individuals with cardiovascular diseases who may not benefit from the increasing use of electronic technological services, such as those who are socioeconomically deprived, lacking Internet access, or having limited eHealth literacy. Further review could help find methods to increase their participation by providing the technical support and technology needed for access to the interventions [12,24]. A broader approach is likely needed to motivate patients to take a more active role in promoting their own health care with the help of technological interventions [13,14]. More information can be obtained by conducting studies with those who own and are comfortable with technology and comparing them with those who have barriers to learning the new methods or appear to be naïve to these interventions [4,5,13]. 

## 5. Conclusions

Overall, from pediatrics to geriatrics, diverse groups and populations can benefit from cardiovascular monitoring through electronic technological interventions. These methods include, but are not limited to, interactive applications on smartphones for monitoring, telehealth, video conferencing, and physical activity trackers [10,11,12,13,14,15,16,17,18,19,20]. Patients mention the benefits of these modalities, which include avoiding obstacles related to medical transportation, adverse climatic conditions, family and work commitments, and the positive aspect of reduced time to complete an appointment [10,11,12,15,16,17,18,19,20]. A one-size-fits-all approach may not work with these modalities and interventions [36,37]. These studies highlight the digital divide as individuals having limited eHealth literacy and lack of technological knowledge are not motivated to participate in these interventions. Generally older age groups (above 69) were much less likely to use mobile technology in all respects, including hesitancy in using apps for health reasons [10,11,42]. The potential use of small-screen devices may be limited by vision challenges and dexterity, which can be remedied by providing larger handheld devices such as tablets [12,24,42]. 

Even though digital health care services show potential for making information widely accessible to patients, it is ironic that this capability may propagate disparities and create obstacles in some scenarios in which these technologies are employed to improve access to health-related information [4,31,32,33,34]. Finding methods to overcome these barriers is important and could be solved to some extent by conducting studies in diverse populations with larger sample sizes in which providing the technology, and the technical support is prioritized [34,35,36,37].

## Figures and Tables

**Table 1 healthcare-12-02055-t001:** Reference table. Characteristics (summary and details) of the articles included.

Article	Sample (*n*)	Type of Study	Intervention and Patient Information	Summary
1. Chow et al. [10]	1424	Randomized control trial (RCT)	1424 patients (mean age = 58 years, 79% male) with heart disease (from 18 Australian teaching hospitals), owning a text-capable mobile phone with the ability to read messages in English, were followed for a year. The participants were compared with a control group with no text messages.	The program delivered consistent education and support to cardiac patients after hopitalization. Results showed favorable response from patients, including high levels of acceptability, usefulness in being a unified source of information, program engagement, and emotional support. However, medication adherence was not improved.
2. Beatty et al. [11]	13	Observational study	13 participants (1 female) mean age = 63, with cardiac surgery, angina, or heart failure, owning a mobile phone or computer with Internet access, participated in this study related to feedback on veteran use of a mobile application. The mobile app VA FitHeart included health education along with reminders and feedback. The app also provided physical activity goal setting, along with daily logs for physical activity tracking and health metric recording (e.g., blood pressure, weight, and mood/emotional well-being).	The study used patient feedback to improve the usability of the app through questionnaitres and semistructured interviews. Patient expectations for using a mobile app for cardiac rehabilitation (CR) included tracking health metrics, introductory training, and sharing data with providers. Patients in the study desired the ability to track physical activity.
3. Harzand et al. [12]	258	Open-label trial	Patients with cardiac disease were required to own an Android or an iOS smartphone in working condition with access to Wi-Fi or a data plan to enroll in this digital health intervention (DHI) program for the cardiac rehabilitation program at the VHA medical center. A total of 258 participants, mean age 60 ± 9 years, 93% male and 48% Black enrolled in the program for three months.	Results indicated that the remote CR with DHI was feasible in the VA hospital setting. Participants’ health status improved with better walking capabilities and low-density lipoprotein cholesterol, while smoking decreased. Additionally, no adverse events were noted.
4. Lunde et al. [13]	113	RCT	113 patients completing cardiac rehab were randomly allocated to the intervention. Mean age of particpants = 59, 22% female (coronary artery disease = 73.4%, 16.8% = valve surgery, and other heart diseases = 9.8%). The intervention consisted of receiving follow-up with the mHealth app or a control group with usual care. Patients were recruited from two CR centers in the eastern region of Norway.	Post-CR patients were compared with control group in this study consisting of individualized follow-up for one year with an app. Improvements were seen in VO2peak, exercise performance, and exercise habits, as well as in self-perceived goal achievement. No other outcomes were different.
5. Villani et al. [14]	268	Descriptive mixed-method study	The questionnaire was distributed among 268 patients attending an outpatient arrhythmia clinic. 82.4% men with mean age 69 years, participated in this study conducted in Northern Italy.	In this study, the results indicate that the patients expressed a greater interest in receiving information related to the effectivenes and integrity of the device. Lower interest toward the clinical status and arrythmic episodes and healthy lifestyle counseling was observed.
6. Kathuria-Prakash et al. [15]	40	Community participatory research design	40 Black women aged 25–45 years with at least two cardiovascular risk factors completed 4 sessions of cardiovascular disease risk reduction education. Additionally, a 6-month smartphone coaching and cardiovascular disease risk reduction monitoring, which targeted heart-healthy lifestyle and behavior modifications, was provided.	The results indicate that the mHealth intervention was a feasible tool for implemeting cardiovascular disease risk reduction for young Black women. Improving the health awareness of the participants had indirect benefits for other family members especially children.
7. Sankaran et al. [16]	32	RCT	The impact of the HeartHab app was studied on 32 patients with coronary artery disease for 4 months in Belgium. Overall, patients’ motivation, physical activity, exercise target achievement, quality of life, and modifiable risk factors were investigated.	Results demonstated positive improvements in VO2 max, glucose, HDL cholesterol, weight, and quality of life.
8. Martin et al. [17]	48	RCT	48 participants (46% women, mean age 58 years) from a cardiology center in Baltimore, owning smartphones, took part in this study for 4 months with the objective that the mHealth intervention with tracking and texting components would increase physical activity.	Smart texts with activity tracking led to the best physical activity outcomes, such as increased daily steps (better outcomes than tracking only).
9. Chen et al. [18]	14	Clinical trial (pilot study)	14 (8–19 year old) patients, at least 1 year postcardiac transplant surgery, underwent a 12–16 week diet and exercise intervention, which was delivered via live video conferencing to improve cardiovascular health.	Results indicate that the lifestyle intervention of exercise and nutrition was feasible with excellent adherence, improvements in cardiac, vascular, nutritional, and functional health.
10. Tierney et al. [19]	24	Cohort study	24 patients with Marfan syndrome (8 to 19 years old) participated in a 6-month physical activity intervention, and their steps were tracked.	Physical activity intervention was feasible in this population and has the potential to decrease the aortic root (AoR) dilation rate.
11. Ziebel, et al. [20]	12	Feasibility study	Mean age of participants was 15.4 years (SD = 3.4) with mean time since cardiac transplant 9.7 years (SD = 4.3). Participants wore a FitBit accelerometer throughout the duration of the study to monitor daily activity levels. The participants underwent the intervention for 16 weeks, which consisted of exercise sessions twice a week for 30 min, supervised by a trained exercise physiologist over a virtual platform. At the conclusion of the intervention, participants repeated the strength and flexibility assessment, a 6MWT, and quality of life (QoL) parameter measurements to compare with baseline.	Results of the study indicate the successful implementaton of a virtual cardiac fitness intervention with improvement in QoL metrics and excellent adherence of participants.

**Table 2 healthcare-12-02055-t002:** Overview of the methodological quality checklist “Critical review form—quantitative studies” [9].

	Chow et al. [10]	Beatty et al. [11]	Harzand et al. [12]	Lunde et al. [13]	Villani et al. [14]	Kathuria-Prakash et al. [15]	Sankaran et al. [16]	Martin et al. [17]	Chen et al. [18]	Tierney et al. [19]	Ziebell et al. [20]
Study purpose: was the purpose stated clearly?	Yes	Yes	Yes	Yes	Yes	Yes	Yes	Yes	Yes	Yes	Yes
Literature: was the relevant and background literature reviewed?	Yes	Yes	Yes	Yes	Yes	Yes	Yes	Yes	Yes	Yes	Yes
Design	RCT	Observational study	Open-label trial	RCT	Descriptive mixed-method study	Community participatory research design	RCT	RCT	Open-label trial	Cohort	Feasibility study
Sample	N = 1424	N = 13	N = 258	N = 113	N = 268	N = 40	N = 32	N = 48	N = 14	N = 24	N = 12
Was the sample described in detail?	Yes	Yes	Yes	Yes	Yes	Yes	Yes	Yes	Yes	Yes	Yes
Was the sample size justified?	Yes	No	No	Yes	No	No	Yes	Yes	Yes	No	No
Results: were results reported in terms of statistical significance?	Yes	Yes	Yes	Yes	Yes	Yes	Yes	Yes	Yes	Yes	Yes
Were the analysis method(s) appropriate?	Yes	Yes	Yes	Yes	Yes	Yes	Yes	Yes	Yes	Yes	Yes
Was clinical importance reported?	Yes	Yes	Yes	Yes	Yes	Yes	Yes	Yes	Yes	Yes	Yes
Were drop-outs reported?	Yes	Yes	Yes	Yes	Yes	Yes	Yes	Yes	Yes	Yes	Yes
Conclusions and clinical implications: were conclusions appropriate given the study methods and results?	Yes	Yes	Yes	Yes	Yes	Yes	Yes	Yes	Yes	Yes	Yes

Abbreviations used in the table: RCT = randomized controlled trial.

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
