# Peer review of "Technological Interventions to Implement Prevention and Health Promotion in Cardiovascular Patients"

_healthcare, 2024, doi:10.3390/healthcare12202055_

Round 1
Reviewer 1 Report
Comments and Suggestions for Authors
(1) Because the “digital divide” is referenced throughout, the Introduction section should provide additional discussion on the topic and the different elements of the divide.
(2) Although many readers will most likely know what MeSH is, it would be helpful to state that it stands for Medical Subject Headings.
(3) Although the authors presented how the 11 articles in their study scored on the critical review form for quantitative studies (in Table 2), they did not explain how the checklist in Table 2 was used. Under what circumstances would an article ‘fail’ using this checklist?
(4) I am concerned about the author’s sample size of 11 articles which discussed studies that took place in different countries. It is difficult to generalize from such a small sample. In addition, it would be difficult to propose solutions when the medical care system and the extent of the digital divide would vary from place to place.
(5) The authors should strengthen their conclusions. There has been more than one study that has argued the Internet could potentially help deliver medical care but differences in Internet access, broadband, digital literacy, etc. are limiting its potential. What have the authors of this article added to this existing literature.
Author Response
Dear Reviewer 1,
Comment: (1) Because the “digital divide” is referenced throughout, the Introduction section should provide additional discussion on the topic and the different elements of the divide.
Reply: The digital divide came apparent after going through the articles in the review and reading about the topic of telehealth in the times of COVID pandemic. A follow up pilot study or commentary article related to this topic is an idea we are considering. Going through the different elements of the divide is a related and significant topic however it is beyond the scope of this review. There is a brief mention in the intro section, which I have added.
Comment: (2) Although many readers will most likely know what MeSH is, it would be helpful to state that it stands for Medical Subject Headings.
Reply: Thank you for your suggestion, I added it to the manuscript.
Comment: (3) Although the authors presented how the 11 articles in their study scored on the critical review form for quantitative studies (in Table 2), they did not explain how the checklist in Table 2 was used. Under what circumstances would an article ‘fail’ using this checklist?
Reply: The articles will fail to be included if they do not meet majority of the criteria, at least 90% -in our case we decided to include all the articles as they met the requirements listed. According to the creators of this tool [9] “The decision as to whether or not to include a study can be made based on meeting a pre-determined proportion of all criteria, or on certain criteria being met. It is also possible to weight certain criteria differently. Decisions about a scoring system or any cut-off for exclusion should be made in advance and agreed upon by all reviewers before critical appraisal commences.”
Also briefly we have mentioned this in the article ( lines 104-106) that the “papers included design, sample size and proper analysis (Table 2); therefore, we included all 11 articles in the review after quality control analysis”.
Comment: (4) I am concerned about the author’s sample size of 11 articles which discussed studies that took place in different countries. It is difficult to generalize from such a small sample. In addition, it would be difficult to propose solutions when the medical care system and the extent of the digital divide would vary from place to place.
Reply: That is mentioned in the limitations section. Majority of the studies were conducted in the USA, with a few studies from Europe (Italy, Norway). The search criteria did not include studies conducted in the USA as an inclusion criteria.
Comment: (5) The authors should enhance their conclusions. There has been more than one study that has argued the Internet could potentially help deliver medical care but differences in Internet access, broadband, digital literacy, etc. are limiting its potential. What have the authors of this article added to this existing literature.
Reply: More content has been added to the conclusion to add strength to the topic of the limitations and digital divide. Our review is unique as we are covering the aspect of diverse cardiac populations and technological interventions. Our team of cardiologists sees multigenerational patients from all over the state, hence our interest is to search the utility of technological interventions implementation in children as well as the VA population of elderly Veterans.
Reviewer 2 Report
Comments and Suggestions for Authors
Thank you for submitting your manuscript on "Technological Interventions to Implement Prevention and Health Promotion in Cardiovascular Patients." Overall, your paper is nicely written and contains some good information on the overlap between cardiovascular disease, and the use of technology in preventing health issues relating from mHealth interventions. A few questions:
1. pg 1, lines 42-44. I would combine these sentences together for clarity, something like, "The use of telecommunications technology to provide health care from a distance, also called telehealth, is very popular and is implemented in medical the care of a variety of chronic diseases, including cardiovascular diseases."
2. pg. 2, lines 48-53. I think you need to specify here that I think you're talking maybe about parents/caregivers monitoring their children...otherwise it sounds like the children (in utero) are using technology to monitor their own cardiovascular health?
4. For Table 1 and 2, is it possible to do horizontal orientation to avoid too much hyphenation of names and results?
5. Pg. 2, line 66. Can you describe more about this "quality control process"?
6. Just curious as to why this study design was chosen (i.e., its benefits) vs. something like systematic review, meta-analysis, etc.
All else looks good!
Author Response
Dear Reviewer 2,
Comment: Thank you for submitting your manuscript on "Technological Interventions to Implement Prevention and Health Promotion in Cardiovascular Patients." Overall, your paper is nicely written and contains some good information on the overlap between cardiovascular disease, and the use of technology in preventing health issues relating from mHealth interventions. A few questions:
1. pg 1, lines 42-44. I would combine these sentences together for clarity, something like, "The use of telecommunications technology to provide health care from a distance, also called telehealth, is very popular and is implemented in medical the care of a variety of chronic diseases, including cardiovascular diseases."
Reply: Thank you for your suggestion. The sentence has been rephrased.
Comment: 2. pg. 2, lines 48-53. I think you need to specify here that I think you're talking maybe about parents/caregivers monitoring their children...otherwise it sounds like the children (in utero) are using technology to monitor their own cardiovascular health?
Reply: This has been added to the paragraph.
Comment: 4. For Table 1 and 2, is it possible to do horizontal orientation to avoid too much hyphenation of names and results?
Reply: I have published similar tables previously and they look good in the printed version. If the editors would like me to change formats of the tables etc. I will be happy to do so.
Comment: 5. Pg. 2, line 66. Can you describe more about this "quality control process"?
Reply: Answer is replied previously to reviewer 1, comment 3. (if needed I can add this to the supplementary file ( as it doesn’t seem to fit within the manuscript).
Comment: 6. Just curious as to why this study design was chosen (i.e., its benefits) vs. something like systematic review, meta-analysis, etc.
Reply: The goal was to write a scoping or narrative review related to diverse populations as we wanted to cover this aspect of diverse age groups of cardiac populations and technological interventions. Our team of cardiologists sees multigenerational patients from all over the state, hence our interest is to search the utility of technological interventions implementation in children as well as following their parents and grandparents if they have congenital or genetic component to their cardiovascular disease. An initial search showed gap in literature related to children with cardiovascular diseases and digital interventions. For such topics these types of reviews yield better results which can contribute to further ideas for exploration related to this topic.
Comment: All else looks good!
Response: Thank you very much!
Reviewer 3 Report
Comments and Suggestions for Authors
The review explores how technological interventions like telehealth and mobile health influence health promotion in cardiovascular patients. It underscores the practicality and acceptance of these technologies while also pointing out a digital divide caused by limited eHealth literacy among certain groups. The review advocates for more inclusive research and strategies to address this gap, emphasizing that customized interventions could enhance cardiovascular outcomes across various populations.
- The introduction provides a solid overview of cardiovascular disease and the role of technological interventions. However, it would be strengthened by explaining why this review is particularly needed. Does it address a gap in the current literature or provide updated insights?
- This section outlines the included studies, showcasing a variety of study designs and interventions. However, the review might be more cohesive if it were structured thematically, such as by types of interventions (e.g., mobile apps, telehealth);
- The discussion section gives a concise summary of the findings, but it could be enhanced by linking these results to the broader context of cardiovascular health management. Additionally, a more critical evaluation of the studies' limitations and a thorough discussion of future research avenues would improve this section;
- The review notes that many studies predominantly involve adult male participants. However, it should delve deeper into how to address this limitation. For example, the discussion could include strategies to better involve underrepresented groups, such as women, the elderly, and those with limited digital literacy. Moreover in order to enrich your discussion the authors could include some specific settings in which telemedicine has been useful such as arrhythmic patients during COVID 19 pandemic: The Feasibility, Effectiveness and Acceptance of Virtual Visits as Compared to In-Person Visits among Clinical Electrophysiology Patients during the COVID-19 Pandemic. J Clin Med. 2023 Jan 12;12(2):620;
- Including an analysis of the wider impacts of these interventions on healthcare systems—such as their potential to reduce hospital readmissions and their cost-effectiveness—would enrich the review;
- While the review touches on aspects of patient motivation and engagement, it could be expanded to delve into the psychosocial effects of using technology for cardiac health management. For instance, how do these interventions affect patients' quality of life, anxiety levels, or mental health?
Comments on the Quality of English Language
authors should try to reduced the overall complexity of sentences in order to improve the readability of their manuscript.
Author Response
Dear Reviewer 3,
Comment: The review explores how technological interventions like telehealth and mobile health influence health promotion in cardiovascular patients. It underscores the practicality and acceptance of these technologies while also pointing out a digital divide caused by limited eHealth literacy among certain groups. The review advocates for more inclusive research and strategies to address this gap, emphasizing that customized interventions could enhance cardiovascular outcomes across various populations.
- The introduction provides a solid overview of cardiovascular disease and the role of technological interventions. However, it would be strengthened by explaining why this review is particularly needed. Does it address a gap in the current literature or provide updated insights?
Reply: Yes the gap in literature is mentioned in these lines-64-68, as shown below (also mentioned in the discussion and conclusion).
Studies generally consist of adult male populations owning smartphones; however cardiovascular disease is present in all age groups regardless of gender, race and socioeconomic status [4,6,7,8]. Additionally, the congenital heart disease population remains underrepresented in many studies [5,6,7]. This review aims to address limitations and explore the status of mHealth strategies, in real clinical and community settings, consisting of diverse populations and demographics.
Comment: - This section outlines the included studies, showcasing a variety of study designs and interventions. However, the review might be more cohesive if it were structured thematically, such as by types of interventions (e.g., mobile apps, telehealth);
Reply: This study is more about the diverse populations, thematic technological analysis is beyond the scope of this narrative review, also limited by the diversity of the technological interventions, cardiac populations and small sample size. That should be a different manuscript covering systematic analysis related to thematic analysis of interventions used in COVID etc, or before and after pandemic.
Comment: - A more critical evaluation of the studies' limitations and a thorough discussion of future research avenues would improve this section;
Reply: The limitations section and future directions section are included in this paper and more content is added to both sections.
Comment: - The review notes that many studies predominantly involve adult male participants. However, it should delve deeper into how to address this limitation. For example, the discussion could include strategies to better involve underrepresented groups, such as women, the elderly, and those with limited digital literacy. Moreover, in order to enrich your discussion the authors could include some specific settings in which telemedicine has been useful such as arrhythmic patients during COVID 19 pandemic: The Feasibility, Effectiveness and Acceptance of Virtual Visits as Compared to In-Person Visits among Clinical Electrophysiology Patients during the COVID-19 Pandemic. J Clin Med. 2023 Jan 12;12(2):620;
Reply: The studies from VA system predominantly involve adult male participants and this limitation has been mentioned. Also addressed these comments previously. The article is not about COVID 19 and cardiology, that is a robust and wide topic that should be addressed in another paper dedicated to just this topic, hence better addressed in future directions. Specifically targeting arrythmia, Kawasaki disease, myocarditis in COVID etc is beyond the scope of the paper.
Comment: - Including an analysis of the wider impacts of these interventions on healthcare systems—such as their potential to reduce hospital readmissions and their cost-effectiveness—would enrich the review;
Reply: This review is related to cardiovascular patients and technological interventions, hence the impact on healthcare systems and cost effectiveness is beyond the scope of this review. It is mentioned as one of the limitations.
Comment: - While the review touches on aspects of patient motivation and engagement, it could be expanded to delve into the psychosocial effects of using technology for cardiac health management. For instance, how do these interventions affect patients' quality of life, anxiety levels, or mental health?
Reply: This would be a great area to cover in another manuscript. The motivation, engagement and emotional factors are mentioned briefly within the review. Overall, this manuscript was related to technological interventions and cardiovascular health, hence anxiety, mental health and psychosocial health is beyond the scope and aims and objectives of this review. That would be an area for the psychologists and psychiatrists to explore.
Comments on the Quality of English Language: authors should try to reduce the overall complexity of sentences in order to improve the readability of their manuscript.
Reply: That is a great point. We have tried to rephrase some of the sentences as mentioned by the previous reviewers.
Round 2
Reviewer 1 Report
Comments and Suggestions for Authors
While I would have preferred a larger sample size, the changes made by the authors were sufficient and appropriate.
Author Response
Appreciate the opportunity and looking forward to seeing the published manuscript, thank you.
Reviewer 3 Report
Comments and Suggestions for Authors
Congratulations to the authors for having answered appropriately to all of my comments
Author Response

(The authors gave the same response as above.)
